# pH-Regulated Strategy and Mechanism of Antibody Orientation on Magnetic Beads for Improving Capture Performance of *Staphylococcus* Species

**DOI:** 10.3390/foods11223599

**Published:** 2022-11-11

**Authors:** Fuying Kang, Yin Yang, Jingwen Li, Erning Chen, Tian Hong, Lulu Zhao, Meihong Du

**Affiliations:** 1Institute of Analysis and Testing, Beijing Academy of Science and Technology, Beijing Center of Physical and Chemical Analysis, Beijing 100094, China; 2Beijing Academy of Science and Technology, Beijing 100094, China

**Keywords:** antibody orientation, pH regulation, immunomagnetic beads, foodborne pathogen, capture efficiency

## Abstract

**Highlights:**

**What are the main findings?**
An pH-regulated strategy can achieve antibody orientation on the surface of magnetic beads.

**What is the implication of the main finding?**
The capture efficiency for *Staphylococcus aureus* of immunomagnetic beads prepared at pH 8.0 was improved.The antibody orientation mechanism was demonstrated using a quantum dots labeled antigen, antigen-binding fragment (Fab) accessibility assay and lysine mimicking.

**Abstract:**

Immunomagnetic beads (IMBs) have been widely used to capture and isolate target pathogens from complex food samples. The orientation of the antibody immobilized on the surface of magnetic beads (MBs) is closely related to the effective recognition with an antigen. We put forward an available strategy to orient the antibody on the surface of MBs by changing the charged amino group ratio of the reactive amino groups at optimal pH value. Quantum dots labeling antigen assay, antigen-binding fragment (Fab) accessibility assay and lysine mimicking were used for the first time to skillfully illustrate the antibody orientation mechanism. This revealed that the positively charged ε-NH_2_ group of lysine on the Fc relative to the uncharged amino terminus on Fab was preferentially adsorbed on the surface of MBs with a negatively charged group at pH 8.0, resulting in antigen binding sites of antibody fully exposed. This study contributes to the understanding of the antibody orientation on the surface of MBs and the potential application of IMBs in the separation and detection of pathogenic bacteria in food samples.

## 1. Introduction

Magnetic beads (MBs) have been extensively used in the fields of biosensors, biomedicine, and biotechnology due to their unique superparamagnetic properties and basic nanostructured characteristics, such as easy functionalization and larger surface to volume ratio [1,2,3,4]. The immunomagnetic beads (IMBs) prepared by immobilizing monoclonal or polyclonal antibodies on the surface of magnetic beads can specifically capture and separate the target and are widely used in the fields of food, hygiene, and environment.

Many studies have reported the potential application of IMBs in the sample pretreatment of the rapid detection of pathogenic bacteria [5,6,7,8]. Although the reaction system of IMS for foodborne pathogens, such as the amount of IMBs, and immunoreaction time has been optimized, the capture efficiency of IMBs is still low [2,3,9], and a high capture performance of target bacteria has become the main goal pursued by researchers [10]. It must be said that the capture and separation efficiency of the target bacteria are closely associated with the accessibility of the Fab of the antibody controlled by its orientation and loading capacity [11,12,13,14,15]. However, the most common strategy for antibodies immobilization is to adopt N-hydroxysuccinimide/1-ethyl-3-(3-dimethylamino) propyl carbodiimide hydrochloride (NHS/EDC) chemistry to activate carboxyl groups on the surface of functionalized solid surface for covalently crosslinking with reactive amine groups of the antibody [16,17,18]. Therefore, covalent coupling strategies do not consider the orientation of the immobilized antibody, and random immobilization of the antibody on solid surface will occur. As a result, the antigen binding sites of the antibody will be blocked and cannot effectively bind the target bacteria, leading to a low capture efficiency.

To date, some effective methods to orient antibodies have been developed, one of which is crosslinking the antibody through carbohydrates groups in the Fc of the antibody with a hydrazine surface [19,20,21]. Additionally, another option is to introduce protein A or protein G, which can specifically bind to the Fc of the antibody [22,23,24,25]. Other strategies involve the use of engineered antibodies by introducing site-specific modifications, such as histidine tags, biotins, and click chemistry reactive groups [26,27,28,29,30]. All of these approaches are based on the immobilization of the antibody through the non-antigen binding Fc region in order to make the Fab more available for antigen recognition. However, the above methods not only affect antibody activity, but also require the recombinant expression of an antibody with further antibody engineering, which is a complex process or involves expensive immobilized proteins. 

It is common knowledge that the basic structure of an antibody contains Fab and Fc. Generally, there are more basic amino acids than acidic amino acids at the Fab terminal of the antibody, i.e., there are more amino groups at the Fab terminal. However, at the Fc terminal of the antibody, the quantity distribution of the two amino acids is more uniform, so the isoelectric point of the Fab terminal is slightly higher than that of the Fc terminal. Such different pH values will lead to an inconsistent charge distribution of antibodies, which will affect the coupling effect of antibodies. In a recent report, the surface electrical properties of the nanomaterial and basic characteristics of the antibody, such as charge distribution and hydrophilicity were considered to develop a simple universal method to orient the antibody [31], suggesting that the pH value of the reaction solution can affect the behavior of an antibody and potentially modulate the directional fixation of antibodies on the surface of nanomaterials [15,32]. Some novel strategies have been proposed: an antibody is first adsorbed into the nanomaterial surface through non-covalent interactions before being coupled to the surface [33,34], which is a process easily controlled by reaction conditions. Other researchers applied an external electric field to orient antibodies on two-dimensional surfaces achieving more than a 100% improvement in the signal-to-noise ratio [35], and a weak electric field in the surrounding environment can affect the interaction between antibodies and surface-charged nanoparticles. Some studies used a model IgG protein (PDB: 1IGY) to investigate the orientation mechanism of antibodies on gold nanoparticles, such as surface amino acid charge distribution and the amount of acid or alkaline amino acid in the Fab or Fc [33,36]. Because different antibody proteins contain different amino acid sequence structures, and each amino acid has its own isoelectric point, which means that different antibodies have different isoelectric points, a slight sequence variation may alter the structure and electrical properties of the antibody surface [37,38,39], thus immobilizing different antibodies on nanomaterials at different pH values (pH 5–8.5), which may lead to different orientations [40,41]. 

Here, we extend the methodology to the orientation of Abs on the surface of carboxylated magnetic beads and its new application fields in food microbiological detection. Based on the amino terminus and lysine side chain amino groups with significantly different pKa values [15], we propose a strategy to control antibody orientation on MBs by modulating the degree of ionization of reactive amino groups. Firstly, the amino terminal of the Fab (pKa = 7.5) and the amino group from the lysine side chain on the Fc (pKa = 10.0) [15] could be regulated with charged properties to achieve the orientation of the Fc, and then N-hydroxysulfosuccinimide (sulfo-NHS) was used to obtain the negatively charged surface of the carboxyl MBs based on the NHS/EDC crosslinking chemistry (Figure 1). 

Considering the prevalence of food contaminated by *Staphylococcus*, we chose the anti-*Staphylococcus* antibody as a representative to prepare the IMBs to validate the feasibility of the strategy. Meanwhile, to probe the mechanism of antibody orientation, we employed a quantum dots labeling antigen, Fab accessibility assay and lysine mimicking to indirectly characterize the behavior of antibodies on the surface of MBs. 

## 2. Methods/Experimental Section

### 2.1. Reagents, Materials, and Apparatus

Carboxyl magnetic beads and preservation solution were obtained from BioMag Scientific Inc. (Wuxi, China), CdSe/ZnS quantum dots (QDs) were purchased from Xingzi New Material Technology Development (Shanghai, China), the *Staphylococcus* monoclonal antibody was obtained from LSBio Inc. (Seattle, WA, USA), goat anti-mouse IgG Fc (anti-Fc) was obtained from ImmunoChemistry Technologies, LLC, (Bloomington, MN, USA), and *Staphylococcus* enterotoxin B (SEB) was purchased from Toxin technology, Inc. (Sarasota, FL, USA). The Micro Bicinchoninic Acid (BCA) Protein Assay Kit, Varioskan Flash Microplate Reader, and DynaMag magnetic separator were purchased from Thermo Fisher Scientific Inc. (Waltham, MA, USA), and the Amino Acid Assay Kit and 2-morpholinoethanesulfonic acid monohydrate (MES) were obtained from Solarbio Science & Technology Co., Ltd. (Beijing, China). Ultra-high temperature instantaneous sterilization milk was obtained from a local supermarket; 1-ethyl-3-(3-dimethylamino) propyl carbodiimide hydrochloride crystalline (EDC), N-hydroxysulfosuccinimide sodium salt (sulfo-NHS), boric acid, sodium tetraborate, N-tert-butyloxycarbonyl(BOC)-L-lysine(ε-NH_2_-lys), and Nε-BOC-L-lysine(α-NH_2_-lys) were purchased from J&K (Beijing, China); Nutrient Broth (NB) and plate count agar (PCA) were purchased from Land Bridge Technology Co., Ltd. (Beijing, China). *S. aureus* (ATCC 25923) was obtained from American Type Culture Collection (ATCC, Manassas, VA, USA), the Zeiss Fluorescence Microscope was purchased from Carl Zeiss Microscopy GmbH (Jena, Germany), QB-628 Rolling Incubator was purchased from Haimen-Kylin-Bell Lab Instruments Co., Ltd. (Nantong, China), and the Zetasizer nano ZS was purchased from Malvern Panalytical Ltd. (Malvern, UK).

### 2.2. Preparation of IMBs

#### 2.2.1. Activation of Carboxyl MBs

Carboxyl MBs (1 mg) were added to a 2 mL low-affinity microcentrifuge tube, washed twice with 1 mL of MEST buffer (0.1 M MES pH 6.0, 0.05% Tween20), and incubated with 10 mg EDC mixed with 10 mg sulfo-NHS (dissolved in MEST buffer) at 25 °C for 30 min. The mixture was then separated by a magnetic separator for 1 min. The supernatant was discarded, and the activated carboxyl magnetic beads (aMBs) were used immediately.

#### 2.2.2. Antibody Immobilization on MBs 

The aMBs (1 mg) were resuspended with 200 μg of anti-*Staphylococcus* antibody [7], which was excessive compared to the previous study in this laboratory, and was dissolved in 0.05 M phosphate-buffered saline containing 0.05% Tween20 (PBST) (pH 6.0 and 7.0) or 0.05 M borate buffer (pH 8.0, 0.05% Tween20) in advance, and then incubated for 2 h at room temperature on a vertical rotating mixer. The activation sites on MBs were blocked by incubation with bovine serum albumin (BSA) for 1 h. The prepared IMBs were washed twice with PBST and stored in preservation solution at 4 °C.

### 2.3. Characteristics of IMBs

#### 2.3.1. Antibody Binding Quantification

The amount of antibody was quantified by measuring the protein concentration with the bicinchoninic acid (BCA) protein quantitation kit. Then, 150 μL of the sample and a 150 μL aliquot of the working reagent were thoroughly mixed and incubated for 2 h at 37 °C. Absorbance at 562 nm was measured with a microplate reader to determine the protein content based on a standard curve. The amount of antibody bound on the surface of the MBs was calculated as the protein concentration difference before and after binding to the MBs.

#### 2.3.2. Size Distribution and Dispersity Characteristics

IMBs and aMBs were prepared at a concentration of 0.33 mg/mL in PBST or different pH buffers, and dispersed by sonication to evaluate the size distribution and zeta potential using the Zetasizer nano ZS instrument. The dispersed states of MBs and IMBs at the same concentration were observed by a fluorescence microscope with 40× objective lens. The anti-*Staphylococcus* antibody was diluted to a final concentration of 60 μg/mL in PBS to measure size distribution.

#### 2.3.3. Capture Efficiency

One hundred micrograms of IMBs was incubated with 10^4^ colony-forming unit (CFU) of *S. aureus* in 1 mL PBST or milk, shaken on a vertical rotating mixer at room temperature for 1 h, and then separated by a magnetic separator. The IMB–bacterial complexes and supernatant were separately diluted to appropriate concentrations and cultured on PCA for 24 h to count the number of bacterial colonies formed. The capture efficiency of the IMBs was equal to the percentage of bacteria captured by IMBs relative to the total bacterial count.

#### 2.3.4. Measurement of the Maximum Binding Capacity

The *S. aureus* suspension cultured overnight at 37 °C in NB was diluted in 1 mL PBST to final concentrations of 10^2^, 10^3^, 10^4^, 10^5^, 10^6^ CFU/mL, and each diluted suspension was incubated with 100 μg of IMBs at room temperature for 1 h, and then separated by a magnetic separator. The capture efficiency of the IMBs was determined by the aforementioned method.

All used reagents, equipment and waste generated in the experiment were sterilized by autoclaving at 120 °C for 1 h to prevent the spread of pathogens. 

### 2.4. Mechanism of Antibody Orientation

#### 2.4.1. Quantum Dots Labeling Antigen Assay

A QDs stock solution (80 μL, 5 mg/mL) was activated with 4 mg EDC and 2 mg sulfo-NHS for 30 min, followed by the addition of 50 μg SEB, and then incubated in the dark for 3 h. The unreacted SEB protein was removed from the system by cryogenic centrifugation and washed twice with PBST. QDs and SEB conjugates were collected and resuspended in 500 μL PBST.

IMBs (50 μg) and the above functionalized QDs (20 μL) were incubated in the dark for 30 min, and then separated by a magnetic separator. The IMB-QD complexes were washed twice with PBST and resuspended in PBST. The obtained supernatant and IMB-QD complexes were used in the subsequent fluorescence analysis.

Confocal fluorescence microscopy imaging was performed on a confocal laser scanning fluorescence microscope, and the IMB-QD complexes were prepared by the aforementioned protocol at the same concentration and measured using the same parameters (with 405 nm of laser and 2 μs of exposure time).

The fluorescence intensity values of the above supernatant and IMB-QD complexes were measured on a microplate reader from 400 to 600 nm with an excitation wavelength of 340 nm.

#### 2.4.2. Fab Accessibility Assay

A mixture of 100 μg IMBs and 10 μg anti-Fc in 200 μL PBS was incubated at 25 °C for 1 h, and the supernatant was collected after separation by a magnetic separator. The amount of anti-Fc was quantified using a BCA protein quantitation kit. The amount of anti-Fc bound to the IMBs was calculated by determining the protein concentration difference between the initial concentration and the concentration in the supernatant according to a standard curve.

#### 2.4.3. Crosslinking Ratio Analysis of ε-NH_2_-lys and α-NH_2_-lys on the Surface of the MBs

The aMBs were incubated with ε-NH_2_-lys and α-NH_2_-lys diluted with 0.05 M PBS (pH 6.0 or 7.0) for 2 h, respectively. The crosslinking ratio was calculated based on the amino acid content before and after crosslinking with MBs. The amino acid content was quantified using an amino acid assay kit based on the principle that the amino group of amino acids can react with indigohydrone to produce blue–violet compounds and measured on a microplate reader at 570 nm.

## 3. Results and Discussion

### 3.1. Characterization of IMBs Properties

To investigate the effect of pH condition on antibody adsorption and orientation, the antibody was pre-ionized at pH 6.0, 7.0, and 8.0, and coupled with MBs to produce pH6-IMBs, pH7-IMBs and pH8-IMBs. A BCA assay was employed to analyze the antibody binding content. The amounts of antibodies bound on the surface of MBs at pH 6.0 and pH 7.0 were higher than the amount of antibody at pH 8.0 (Figure 2A), which is consistent with the recommended lower pH in NHS/EDC coupling protocol. We speculated that the surface of negatively charged carboxyl magnetic beads would adsorb more antibodies because activated amino groups on the antibodies were positively charged below pH 7.5. Furthermore, we analyzed the size distribution of the antibody, the MBs, and the IMBs. The observed size distribution of IMBs was larger than that of the sum of the MB and antibody size due to the presence of monomers (16 nm) and polymers (79 nm) of the *Staphylococcus* antibody (Figure 2B,C). Finally, the dispersity of the aMBs and IMBs was characterized by microscopy imaging and the zeta potential measurement, taking into account its influence on the antibody binding capacity and target recognition [42]. The results show that MBs exhibited a better dispersion and were more negatively charged compared to the IMBs due to the antibody coating and blocking with BSA [4], while there was no significant difference between IMBs at pH 6.0–8.0 (Figure 3, Table 1). Based on the above data of the binding amount and dispersion of the IMBs, the higher capturing capacity of pH6-IMBs, pH7-IMBs and pH8-IMBs on the target antigen would ideally be taken for granted, following to verify this inference via a capture efficiency measurement.

### 3.2. Capture Efficiency of pH6-IMBs, pH7-IMBs and pH8-IMBs

The capture efficiency on *S. aureus* of pH6-IMBs, pH7-IMBs and pH8-IMBs with equal amounts was measured in PBS or milk. The results show that the capture efficiency of pH8-IMBs was higher than that of pH6-IMBs or pH7-IMBs (Figure 4A). The maximum capacity of the binding antigen of pH6-IMBs, pH7-IMBs and pH8-IMBs was 10^5^ CFU (Figure 4B); meanwhile, at 10^3^–10^6^ CFU/mL of *S. aureus*, pH8-IMBs have certain advantages in capture efficiency compared to pH6-IMBs and pH7-IMBs. However, at 10^2^ CFU/mL of the *S. aureus*, the capture efficiency of pH8-IMBs was not significantly higher than that of pH6-IMBs and pH7-IMBs as the amount of used IMBs exceeded the actual amount needed. Furthermore, we investigated the capture efficiency of pH6-IMBs pH7-IMBs and pH8-IMBs with different amounts on 10^0^ CFU/mL of *S. aureus*. The results show that the capture efficiency of all the IMBs was 100% when 20 or 50 μg of IMBs were used, indicating that the amount of used IMBs was excessive. While the amount of IMBs was reduced to 10 μg, the capture efficiency on target bacteria of pH8-IMBs was significantly higher than that of pH6-IMBs or pH7-IMBs (Figure 4C,D). The above results show that the antibody binding capacity of pH8-IMBs was lower, while the capture efficiency was higher than that of pH6-IMBs or pH7-IMBs, and these differences were not related to the dispersity. We speculate that the Fab fragment that contains the antigen binding site was in a different exposure status at pH 6.0, 7.0, and 8.0 during antibody immobilization, and the more the Fab fragment was exposed, the more the antigen was recognized.

### 3.3. Fluorescence Analysis of Antibody Orientation on MBs

To visualize target antigen captured and determine the orientation of the antibody on the surface of MBs, we taken an approach: antigen SEB labeled by QD (QD-SEB) bound to the surface of IMBs to form the IMB-QD complexes by a further immune response step, and the fluorescence intensity of IMB-QDs were analyzed after magnetic separation, which should be related to the amount of antigen captured and exposure status of the Fab fragment (Figure 5A). First, we measured the excitation and emission spectrum of QD-SEB conjugates and QDs by microplate reader. After coupling with the SEB, the maximum excitation and emission wavelength of QD-SEB complexes was slightly red-shifted compared to QDs due to the surrounding organic layer (antigen) [43] (Figure 5E). Additionally, the fluorescence microscopy imaging visually indicated that pH6-IMBs, pH7-IMBs and pH8-IMBs can recognize and capture QD-SEB to form IMB-QD complexes (Figure 5B–D). The fluorescence intensity in the supernatant (pH6-S-QD, pH7-S-QD and pH8-S-QD) was from the unbound QD-SEB complexes after pH6-IMBs, pH7-IMBs and pH8-IMBs capture and separation. These results show that the fluorescence intensity of pH8-IMB-QD complexes is higher than pH6-IMB-QD or pH7-IMB-QD complexes, and that of the corresponding supernatant is lower than that of pH6-S-QD and pH7-S-QD. The above experiments confirmed that pH8-IMBs possessed more exposed Fab fragments and more antigen recognition sites to bind more antigen molecules than pH6-IMBs and pH7-IMBs (Figure 5F,G).

### 3.4. Fab Accessibility Analysis on IMBs

In order to obtain more direct evidence that the Fab fragment of the pH8-IMBs is more exposed than in the pH6-IMBs or pH7-IMBs, and based on the fact that anti-Fc antibody can specifically recognize the exposed Fc region of the IMBs, we evaluated the accessibility of Fab on IMBs by determining the amount of anti-Fc antibody bound to IMBs. Figure 6A reveals that higher pH values would result in fewer anti-Fc antibodies bound to IMBs, indicating that the Fc region of the *Staphylococcus* antibody was more attached to the surface of the MBs at pH 8.0. In other words, the Fab fragment of pH8-IMBs was more exposed to the antigen, which increased the likelihood of IMBs of recognizing *S. aureus*, resulting in higher capture efficiency. 

### 3.5. Lysine Mimicking

Considering the complex microenvironment in which lysine is located, we only chose the representative ε-NH_2_-lys (α-NH_2_ is protected by a BOC protecting group) and α-NH_2_-lys (ε-NH_2_ is protected by a BOC protecting group) to mimic the ε-NH_2_ of lysine and the amino terminal of the antibody, respectively, in order to accurately understand the mechanism of the oriented immobilization of antibody. We studied the coupling ratio of aMBs with α-NH_2_-lys and ε-NH_2_-lys at pH 6.0, 7.0 and 8.0. Although the data of the coupling rate were not obtained due to the poor solubility of amino acids at pH 8.0, the results also show that the coupling ratio of α-NH_2_-lys significantly decreased compared to that of ε-NH_2_-lys at a higher pH value (Figure 6B). The positively charged α-NH_2_-lys and ε-NH_2_-lys could couple with negatively charged aMBs at pH 6.0, whereas only the positively charged ε-NH_2_-lys could couple with aMBs at pH 7.0. Similarly, at pH 8.0, the ε-NH_2_ group of lysine residues in the Fc region was more prone to being adsorbed in aMBs relative to the amino terminus of the antibody, which contributed to the oriented immobilization of the Fc fragment on the surface of MBs, thereby fully exposing the Fab fragment and enabling the recognition of more antigens.

## 4. Conclusions

In this study, we proposed an effective approach to control antibody orientation on the surface of MBs by modulating the degree of ionization of reactive amino groups. The mechanism of oriented immobilization of antibody on MBs was studied through the use of a QDs labeled antigen, Fab accessibility assay and lysine mimicking. MBs activated by EDC/sulfo-NHS were negatively charged and adsorbed the positively charged amino groups of the antibody (amino terminus and ε-NH_2_ of lysine) through electrostatic interactions before crosslinking with the MBs. The amino terminus of the Fab fragment and the ε-NH_2_ of lysine from Fc region were both positively charged at pH < 7.5, leading to the random immobilization of the antibody on MBs. At pH > 7.5, the positively charged ε-NH_2_ group of lysine was preferentially adsorbed on the negative surface relative to the uncharged amino terminus, resulting in Fc-oriented immobilization and a more exposed Fab fragment (Figure 6C). As a result, we improved the capture efficiency of IMBs on *S. aureus* in PBS or milk by controlling the orientation of *Staphylococcus* antibodies immobilized on MBs at pH 8.0, and higher capture efficiencies can be achieved with a lower amount of pH8-IMBs. This strategy would be very useful for the preparation of IMBs for effectively capturing and separating pathogenic bacteria from a complicated food matrix with lower cost, thereby improving specificity and sensitivity integrated with PCR, enzyme-linked immunosorbent assay (ELISA), luminescence and electrochemistry for the detection of trace pathogenic microorganisms. 

## Figures and Tables

**Figure 1 foods-11-03599-f001:**
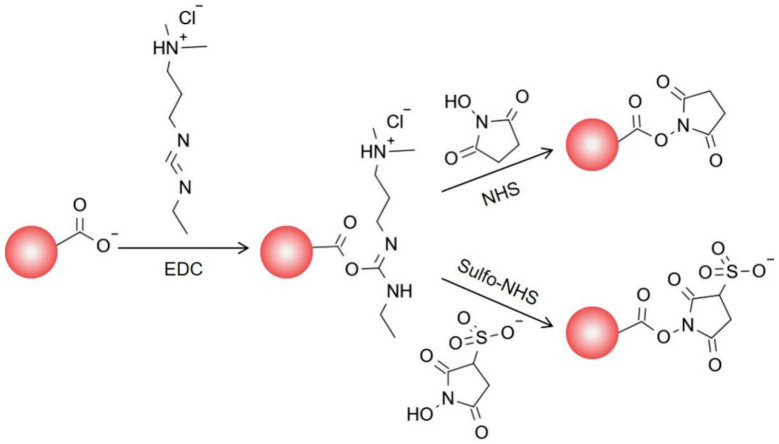
Scheme of activation mechanism of carboxyl MBs based on NHS/EDC chemistry.

**Figure 2 foods-11-03599-f002:**
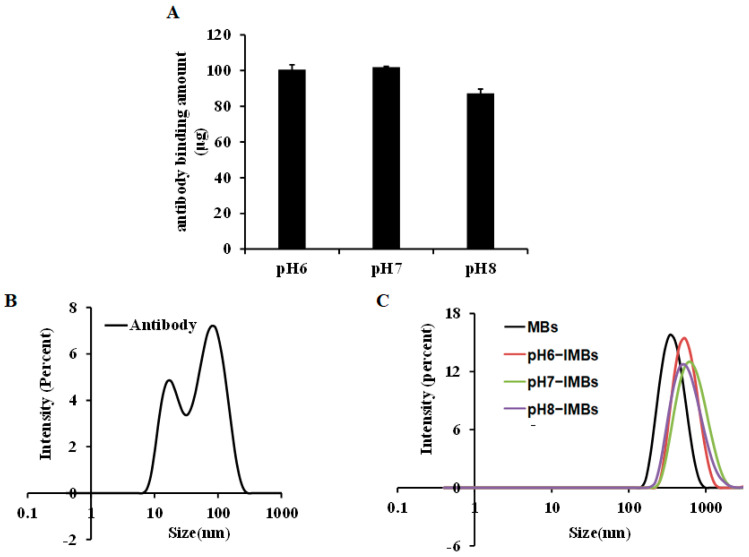
The characterization of IMBs properties. (**A**) Amount of antibody binding to MBs at pH 6.0, 7.0 and 8.0, (**B**) the size distribution assays of antibody, (**C**) and the size distribution assays of MBs, pH6−IMBs, pH7−IMBs, and pH8−IMBs.

**Figure 3 foods-11-03599-f003:**
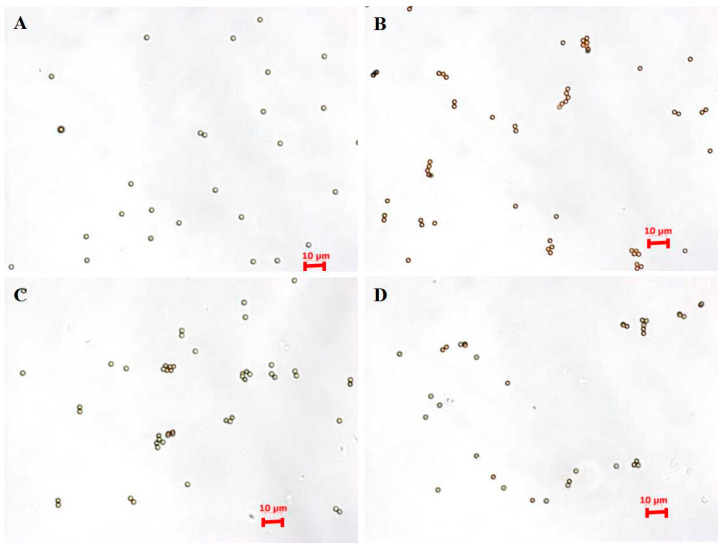
Microscopy images of MBs (**A**) pH6-IMBs (**B**) pH7-IMBs (**C**) and pH8-IMBs (**D**).

**Figure 4 foods-11-03599-f004:**
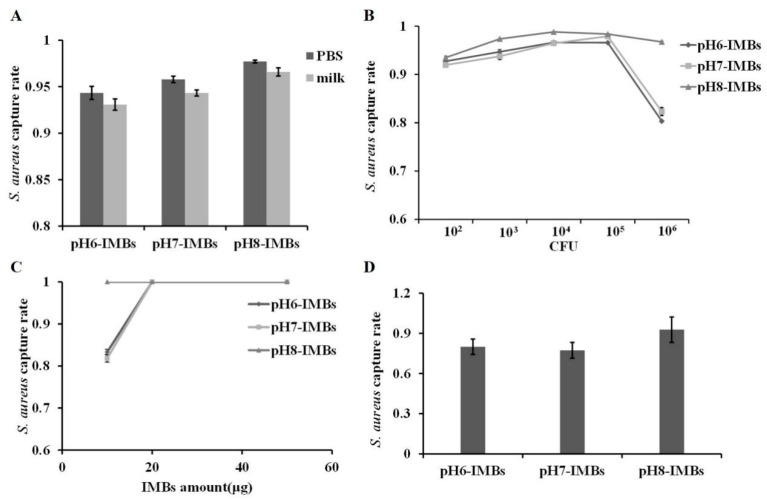
Capture efficiency on *S. aureus* of pH6-IMBs, pH7-IMBs, and pH8-IMBs. (**A**) The capture efficiency of 100 μg IMBs on *S. aureus* in PBS and in milk, (**B**) Maximum binding capacity analysis of 100 μg IMBs, (**C**) The capture efficiency of IMBs with 10, 20 and 50 μg on 10^0^ CFU/mL of *S. aureus*, (**D**) The capture efficiency of 10 μg of IMBs on 10^0^ CFU/mL of *S. aureus*.

**Figure 5 foods-11-03599-f005:**
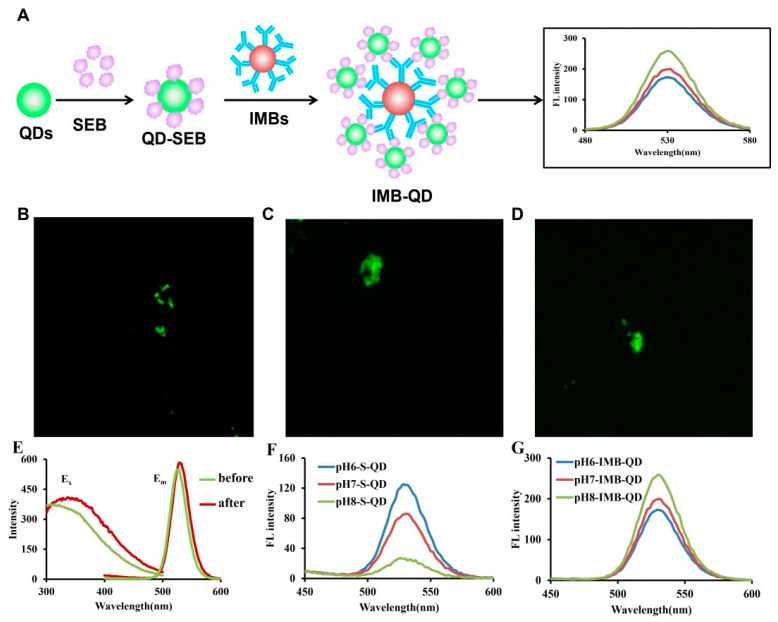
Fluorescence analysis of antibody orientation on MBs. (**A**) Schematic represent for fluorescence analysis, fluorescence microscopy images of pH6-IMB-QD (**B**), pH7-IMB-QD (**C**) and pH8-IMB-QD (**D**), the excitation and emission spectra of QDs and QD-SEB complexes (**E**), fluorescence (FL) intensity analysis of QDs in the supernatant (**F**) and IMB-QD complexes (**G**).

**Figure 6 foods-11-03599-f006:**
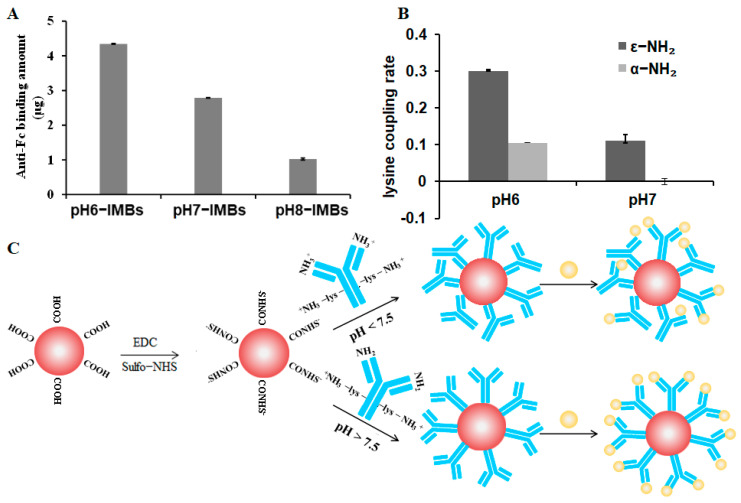
Mechanism of antibody orientation. (**A**) Amount of anti−Fc antibody bound to IMBs, (**B**) The coupling ratio assay of α−NH_2_−lys and ε−NH_2_−lys at pH 6.0 and 7.0, (**C**) The proposed mechanism of antibody orientation on MBs for *S. aureus.* Red spheres represent MBs, yellow spheres represent *S. aureus*, and blue stick structures represent antibody.

**Table 1 foods-11-03599-t001:** The dispersity of aMBs and IMBs.

Sample	Zeta Potential (mV)
aMBs-pH6	−47.73
aMBs-pH7	−41.87
aMBs-pH8	−57.07
pH6-IMBs	−15.33
pH7-IMBs	−17.23
pH8-IMBs	−15.83

## Data Availability

All data and materials during the current study are available from the corresponding author on reasonable request.

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
