# Peer review of "pH-Regulated Strategy and Mechanism of Antibody Orientation on Magnetic Beads for Improving Capture Performance of Staphylococcus Species"

_foods, 2022, doi:10.3390/foods11223599_

Round 1

Reviewer 1 Report

In this manuscript, authors optimized the conjugation efficiency according to pH. They proposed the mechanism of high antigen-binding efficiency at higher than pH 7.5 was because of the electrostatic interactions. By using antibodies conjugated magnetic beads, authors captured S. aureus in PBS and milk. The topic of this manuscript is well fit to the Foods and the logic of manuscript is scientifically sound. Their strategy to analysis of orientation of conjugated antibodies was also technically sound. I would recommend to publish this article after minor modification.

-      In all figures, symbols are too small to distinguish.

Author Response

Thank you very much for your helpful comments. According to your suggestion, we have modified the unclear symbols in the figures, especially the all ruler symbols in Figure 4.

Reviewer 2 Report

Kang and co-authors proposed a strategy to control the antibody orientation on magnetic beads (MBs) to produce immunomagnetic beads (IMBs) with improved antigen capture efficiency. They activated carboxyl magnetic beads using EDC-NHS chemistry, and also prepared the anti-Staphylococcus antibody in three buffer solutions with different pH values (6.0, 7.0, and 8.0). Subsequently, the three prepared antibodies were separately immobilized on the activated carboxyl magnetic beads to obtain three types of IMBs (pH6-IMBs, pH7-IMBs, and pH8-IMBs). The capture efficiency of the three obtained IMBs was investigated in PBS and milk. Finally, the authors exploited quantum dots labeling antigen assay, Fab accessibility assay, and Lysine mimicking to observe that pH8-IMBs has the highest antigen capture efficiency among the three produced IMBs. However, there is a lack of discussion on the main motivation of the present manuscript. The rationale is unclear that the orientation could improve the capture efficiency, and there is no other studies introduced as the control to underpin this hypothesis. Moreover, the EDS-NHS chemistry has been widely used in literature to activate carboxyl MBs (Biosensors and Bioelectronics 43, 274–280, Nanoscale, 2015,7, 3768-3779, etc.). Also, the impact of pH on antibody orientation has already been investigated in Bioconjugate Chem. 2019, 30, 4, 1182–1191, although the type of particles and pH range covered in the present manuscript is different. Other comments are listed below:

1.       The introduction should be more informative about this study. For example, the main focus of the present study is to investigate the effect of the pH solution in which the antibody is dissolved, and all the results are based on these pH differences. However, the introduction is poorly written in which the different pH scenario is not discussed.

2.       As a reason for the higher capture efficiency of pH8-IMBs despite lower antibody binding than pH6-IMBs and pH7-IMBs, the final three lines of page 7 state: “We speculate that the Fab fragment which contains the antigen binding site was in a different exposure status at pH 6.0, 7.0, and 8.0 during antibody immobilization, and the more the Fab fragment was exposed, the more the antigen was recognized.” If the experiment conditions during antibody immobilization for three pH are not similar, other results of this study would also be questionable.

3.       As a justification for more Fab accessibility of pH8-IMBs, both the abstract and the last line of page 10 states: “At pH > 7.5, the positively charged ε-NH2 group of lysine was preferentially adsorbed on the negative surface relative to the uncharged amino terminus, resulting in Fc oriented immobilization and more exposed Fab fragment.” The opinion that the amino terminus of Fab fragment is uncharged at pH>7.5 is not investigated, and no reason is given to justify it.

4.       In line 4 of the abstract, the word “through” has been typed wrong.

Author Response

Thank you very much for your helpful comments. In the preface of the manuscript, the discussion of the research objectives was supplemented and rationale that the orientation of antibody can improve the capture efficiency was described, meanwile, some valuable references have also added accordingly at the appropriate position.

Responses to other comments are listed below:

  1. The introduction should be more informative about this study. For example, the main focus of the present study is to investigate the effect of the pH solution in which the antibody is dissolved, and all the results are based on these pH differences. However, the introduction is poorly written in which the different pH scenario is not discussed.

Response: We have modified the introduction, added the principle of pH-regulated antibody immobilization, and discussed the research results and problems of different regulation strategies. Please see the detailed description in manuscript. Thank you!

  1. As a reason for the higher capture efficiency of pH8-IMBs despite lower antibody binding than pH6-IMBs and pH7-IMBs, the final three lines of page 7 state: “We speculate that the Fab fragment which contains the antigen binding site was in a different exposure status at pH 6.0, 7.0, and 8.0 during antibody immobilization, and the more the Fab fragment was exposed, the more the antigen was recognized.” If the experiment conditions during antibody immobilization for three pH are not similar, other results of this study would also be questionable.

Response: Thank you for your question. In this study, the influence of pH value is mainly focused to study on the immobilization behavior of antibodies on magnetic beads. Here, the parameter of pH value is the only variable in the experiment, and other experimental conditions are the same, from which the results also come.

  1. As a justification for more Fab accessibility of pH8-IMBs, both the abstract and the last line of page 10 states: “At pH > 7.5, the positively charged ε-NH2 group of lysine was preferentially adsorbed on the negative surface relative to the uncharged amino terminus, resulting in Fc oriented immobilization and more exposed Fab fragment.” The opinion that the amino terminus of Fab fragment is uncharged at pH>7.5 is not investigated, and no reason is given to justify it.

Response: Thank you for your question. Referring to the study (Sara Puertas,Biosensors and Bioelectronics 43 ,2013), the α-NH2 group of amino terminal of Fab presents pKa 7.5, and so, at pH>7.5, the α-NH2 is not ionized and shows uncharged state, which was supported by the experiment results of Fab accessibility of pH8-IMBs in our study.

  1. In line 4 of the abstract, the word “through” has been typed wrong.

Response:The wrong word has been corrected in the abstract. thank you!

Reviewer 3 Report

The authors present a study for pH-dependent conjugation of specific antibodies to magnetic beads using the NHS/EDC coupling method. Three different pH were studied for coupling efficiency and capture target efficiency. Although the coupling procedure is well known, as well as that is also pH dependent, the authors conclude that pH also has an effect on the orientation of the antibody, so as the Fab fragment is more available for target capturing. The authors have documented their experiments very well and clearly presented their results. However, the pH 9.0 should also be included in the study to ensure that pH 8.0 is the optimum pH for the coupling reaction.

Moreover, as the optimum pH for the coupling to the beads depends on the charge of the antibody used, the title has to be more specific. An alternative title may be: pH-regulated strategy and mechanism of antibody orientation on magnetic beads for improving capture performance of Staphylococcus species.

The following publication should also be added:

Luciani et al. 2016. Rapid Detection and Isolation of Escherichia coli O104:H4 from Milk Using Monoclonal Antibody-coated Magnetic Beads. Frontiers in Microbiology 7: 942.

The authors should clearly state what that is the extra information of their manuscript compared to this publication.

Author Response

Thank you very much for your helpful comments. The factors affecting the capture efficiency of immunomagnetic beads include two aspects: on the one hand, the coupling amount of antibodies on the surface of magnetic beads; On the other hand, it is the proportion of antibody fixed on the surface of magnetic beads. In the process of antibody coupling with NHS-EDC, in which the coupling is generally activated well at pH5.0-8.0, and the amount of antibody coupling will decrease with the increase of pH value. Therefore, at pH9.0, even though the antibody theoretically belongs to directional immobilization, the amount of antibody coupling will decrease rapidly , and its negative impact on the capture rate will far exceed that of directional immobilization. Therefore, the pH 9.0 had not been considered to be included in the study.

According to your suggestion, we revised that title to “pH-regulated strategy and mechanism of antibody orientation on magnetic beads for improving capture performance of Staphylococcus species”. Thank you!

Moreover, this study (Luciani et al. 2016. Frontiers in Microbiology 7: 942.) added to the forefront of our manuscript is worthy of reference in preparing immunomagnetic beads with monoclonal antibodies to get good capturing ability and to achieve rapid detection O104:H4 from Milk, and expend the application of immunomagnetic beads in food. Thank you!

Round 2

Reviewer 2 Report

The authors have addressed my previous comments.

Reviewer 3 Report

The authors have confronted wll with the suggestions.